# Evaluation of the efficacy and safety of conventional and interlaminar full-endoscopic decompressive laminectomy to treat lumbar spinal stenosis (ENDO-F trial): Protocol for a prospective, randomized, multicenter trial

**Jin-Sung Kim**[ID][1], **Junseok Bae**[2], **Dong Chan Lee**[3], **Sang-Ha Shin**[2], **Han Joong Keum**[2], **Young Soo Choi**[2], **Sang Soo Eun**[4], **Seung Ho Shin**[3], **Hyun Jin Hong**[3], **Ji Yeon Kim**[3], **Tae Hyun Kim**[3], **Woojung Lim**[1], **Junghoon Kim**[1], **Sang-Min Park**[5], **Hyun-Jin Park**[6], **Hong-Jae Lee**[ID][7] *

1 Department of Neurosurgery, Seoul St Mary's Hospital, College of Medicine, The Catholic University of Korea, Seoul, Korea, 2 Department of Neurosurgery, Chungdam Wooridul Spine Hospital, Seoul, Korea, 3 Department of Neurosurgery, Wiltse Memorial Hospital, Anyang, Korea, 4 Department of Orthopaedic Surgery, Chungdam Wooridul Spine Hospital, Seoul, Korea, 5 Spine Center and Department of Orthopaedic Surgery, Seoul National University College of Medicine and Seoul National University Bundang Hospital, Seongnam, Korea, 6 Department of Orthopedic Surgery, Spine Center, Kangnam Sacred Heart Hospital, Hallym University College of Medicine, Seoul, Korea, 7 Department of Neurosurgery, Daejeon St Mary's Hospital, College of Medicine, The Catholic University of Korea, Seoul, Korea

* kosailee73@gmail.com

**Data Availability Statement:** The data collected will be anonymized and uploaded to the iCReaT

## Abstract

Lumbar spinal stenosis is a common spinal degenerative condition. Minimally invasive interlaminar full-endoscopic decompressive laminectomy provides greater patient satisfaction and faster recovery than open decompressive laminectomy. The aim of our randomized controlled trial will be to compare the safety and efficacy of interlaminar full-endoscopic laminectomy and open decompressive laminectomy. Our trial will include 120 participants (60 per group) who will undergo surgical treatment for lumbar spinal stenosis. The primary outcome will be the Oswestry Disability Index measured at 12 months postoperatively. Secondary patient-reported outcomes will include back and radicular leg pain measured via a visual analog scale; the Oswestry Disability Index; the Euro-QOL-5 Dimensions score measured at 2 weeks and at 3, 6, and 12 months postoperatively; and patient satisfaction. The functional measures will include time to return to daily activities postoperatively and walking distance/time. The surgical outcomes will include postoperative drainage, operation time, duration of hospital stay, postoperative creatine kinase (an indicator of muscle injury) level, and postoperative surgical scarring. Magnetic resonance and computed tomography images and simple radiographs will be obtained for all patients. The safety outcomes will include surgery-related complications and adverse effects. All evaluations will be performed by a single assessor at each participating hospital who will be blinded to group allocation. The evaluations will be conducted preoperatively and at 2 weeks and 3, 6, and 12 months postoperatively. The randomized, multicenter design of the trial, blinding, and justification of the sample size will reduce the risk of bias in our trial. The results of the trial will provide data

system, though this system is not accessible to the public. Access to the data set will be provided to the Data Management Committee of the Korean Government Research Consortium. The study results will be published in a peer-reviewed journal.

**Funding:** This research is supported by a grant from the Korea Health Technology R&D Project through the Korea Health Industry Development Institute, funded by the Ministry of Health & Welfare, Republic of Korea (grant number: HC20C0163). The funder did not provide support in the form of salaries for any author and did not have any additional role in the study design, data collection and analysis, decision to publish, or preparation of the manuscript. No additional external funding was received for the study.

**Competing interests:** Jin-Sung Kim works as a consultant for RIWOSpine, GmbH, Germany, Stöckli Medical AG, Switzerland, and Elliquence, LLC, USA. Junseok Bae is a consultant for Joimax, GmbH, Germany. This does not alter our adherence to the PLOS ONE policies on sharing data and materials.

regarding the use of interlaminar full-endoscopic laminectomy as an alternative to open decompressive laminectomy that results in similar surgical findings with less invasiveness.

**Trial registration:** This trial is registered at cris.nih.go.kr. (KCT0006198; protocol version 1; 27 May 2021).

## Introduction

Lumbar spinal stenosis is characterized by a narrowing of the spinal canal with subsequent compression of the thecal sac and possible compression of individual rootlets due to facet joint compression and ligament hypertrophy. Lumbar spinal stenosis is a common spinal degenerative disease among elderly individuals, and surgical treatment is often indicated [1]. The prevalence of lumbar spinal stenosis based on clinical and radiological diagnoses has been reported to be up to 39% [2, 3]. Conventional open laminectomy is considered the gold standard surgical treatment for lumbar spinal stenosis and requires the dissection and detachment of muscles to expose and resect the lamina and ligaments, which can lead to injuries to the soft tissues and paraspinous muscles [4–7]. However, the development of the tubular retractor and the use of microscopes and endoscopes have rendered minimally invasive surgery an alternative surgical approach for spinal decompression [1, 4, 8–12]. Percutaneous interlaminar full-endoscopic laminectomy is a minimally invasive technique that allows for a smaller skin incision; lesser muscle damage, scarring, blood loss, and pain; and a shorter recovery than conventional open laminectomy [5, 13–18]. Previous studies have reported that the clinical outcomes of interlaminar full-endoscopic laminectomy are not different from those of conventional laminectomy [13–15, 19–22]. However, all previous studies comparing interlaminar full-endoscopic laminectomy and conventional open laminectomy are retrospective or single-center prospective studies [4, 13–15, 20, 21, 23–25]; therefore, a multicenter, randomized controlled trial (RCT) is needed.

We have designed a multicenter RCT to compare the outcomes of interlaminar full-endoscopic laminectomy and conventional open laminectomy. We hypothesize that the efficacy and safety of interlaminar full-endoscopic laminectomy and conventional open laminectomy for the lumbar spine will be similar.

## Materials and methods

### Ethics

The methods of our study protocol were approved by the institutional review boards of the three participating hospitals, i.e., the Catholic University of Korea Seoul St. Mary's Hospital, KC21ENDI0239B; Cheongdam Wooridul Spine Hospital Seoul, 2121-07-WSH-009; and Wiltse Memorial Hospital, 2021-W04, on July 06, 2021, July 08, 2021, and September 08, 2021, respectively. All future protocol modifications that may affect the research data will be approved by the research ethics committees prior to implementation. Informed consent regarding the surgery and the use of research data will be obtained from each participant, who meets the eligibility criteria, prior to enrollment in the trial.

### Trial design

The aim of this study is to evaluate the non-inferiority of the outcomes of interlaminar full-endoscopic laminectomy versus open decompressive laminectomy by using a multicenter

RCT design. In this multicenter RCT, blinding of the healthcare providers, who assess the participants and who have been approved by the institutional review boards of all the three aforementioned participating hospitals, will be performed.

## Study cohort

The trial will include 120 adults aged 20–80 years with lower extremity radiculopathy due to lumbar spinal stenosis. A total of 60 participants recruited from the three participating hospitals will be allocated to the control group (open laminectomy), and 60 participants will be allocated to the intervention group (interlaminar full-endoscopic laminectomy). The SPIRIT schedule of enrollment, interventions, and assessments is shown in Fig 1.

## Inclusion criteria

Patients who have failed conservative treatment and are considered suitable for decompression surgery; those aged 20–80 years with ≥ lumbar spinal stenosis grade B based on Schizas classification [26]; and those with lumbar central canal stenosis meeting the above criteria who will agree to undergo one to two segment posterior spinal decompression surgeries, will be able to follow instructions and provide consent for research participation, and are willing to participate and fully comply with the follow-up protocol will be included in this trial.

| Visit type | Screening | Surgical procedure | Follow-up | | | |
|---|---|---|---|---|---|---|
| Visit | 1 | 2 | 3 | 4 | 5 | 6 |
| Visit week | -4 -0 weeks | 0-2 days | 2 weeks ± 5 days | 12 weeks ± 4 weeks | 24 weeks ± 8 weeks | 52 weeks ± 8 weeks |
| Informed consent | ■ | | | | | |
| Demographics* | ■ | | | | | |
| Inclusion / Exclusion | ■ | | | | | |
| Randomization | | ■ | | | | |
| Operation | | ■ | | | | |
| MRI (or CT)† | ■ | ■ | | | | |
| Simple radiographs | ■ | | ■ | ■ | ■ | ■ |
| ODI | ■ | | ■ | ■ | ■ | ■ |
| EQ-5D-5L | ■ | | ■ | ■ | ■ | ■ |
| VAS | ■ | | ■ | ■ | ■ | ■ |
| POSAS | | | | ■ | ■ | ■ |
| Other surveys‡ | | | ■ | ■ | ■ | ■ |
| Adverse events | | ■ | ■ | ■ | ■ | ■ |

**Fig 1. Schedule of enrollment, interventions, and assessments.** Abbreviations: MRI, magnetic resonance imaging; CT, computed tomography; ODI, Oswestry Disability Index; EQ-5D-5L, EuroQol 5 Dimension 5 level; VAS, visual analog scale; POSAS, Patient and Observer Scar Assessment Scale. * Baseline patient characteristics, including data about past medical/surgical history, physical examination, and laboratory tests. † CT will be performed when MRI cannot be performed. ‡ including satisfaction with the surgery, walking distance/time, and time required to return to daily activities postoperatively.

## Exclusion criteria

Participants with spondylolisthesis (Meyer grade $\geq$ II); lumbar disc herniation of root compression grade $\geq$ II based on Park's classification [27]; a history of lumbar spinal surgery at the same level as would be performed during the trial; degenerative lumbar scoliosis (Cobb angle > 20˚); other spinal diseases (such as ankylosing spondylitis, spinal tumor, spinal fracture, or neurologic disorders); or psychological disorders (such as dementia, intellectual disability, or drug abuse) will be excluded from the trial. Participants who are considered unfit for the study by their physicians will also be excluded (patients who are sensitive to pain, have myofascial pain syndrome, previous paresis, or severe knee osteoarthritis).

## Enrollment

Patients who agree to proceed with one to two segment posterior decompression surgeries for lumbar spinal stenosis at each of the three participating hospitals will be recruited for participation in this trial. Social media will not be used to recruit participants in this trial. Researchers at each of the three participating hospitals will screen potential participants to determine their eligibility, and assessors blinded to the participants' personal information will perform baseline testing at each of the three hospitals after informed consent is obtained; this will include magnetic resonance imaging (MRI) findings, simple radiographs, Oswestry Disability Index (ODI) score, EuroQol-5-dimension-5-level (EQ-5D-5L) questionnaire, and visual analog scale (VAS). The questionnaires and clinical and radiological analyses conducted in this trial will not harm the study participants.

## Randomization and follow-up

After the baseline assessment, the participants will be randomly assigned into the control (open laminectomy) or intervention (interlaminar full-endoscopic laminectomy) group using a permuted blocked randomization method. The randomization list will be computer-generated, and the randomization code will be sent to the surgeons in each participating hospital using consecutively numbered opaque envelops 12 h prior to the surgery. This central allocation will be conducted by a contract research organization (Helptrial). The randomization lists of the three participating hospitals will be integrated into a web-based electronic case report form (eCRF) platform (internet-based clinical research and trial (iCReaT), icreat.nih.go.kr). Follow-up assessments will be performed by an independent assessor at each site at 2 weeks and at 3, 6, and 12 months postoperatively (Fig 2).

## Assessments and blinding

All assessments will be performed by one assessor at each of the three participating hospitals. The assessors will be blinded to the participants' group allocations. The VAS score of back and leg pain, EQ-5D-5L, and ODI are patient-reported, but these scorings will be obtained with blinded assessor's assistance. Postoperative surgical scarring, extent of disc removal, and injury to the facet joint will be measured by a blinded assessor using postoperative magnetic resonance (MR) or computed tomography (CT) images. Complications on simple radiographs, adverse events, and surgery-related events will be evaluated and managed by a blinded assessor. The assessor at each of the three participating hospitals will attend a training session prior to data collection to ensure consistency among the sites. The surgeon will have knowledge of the procedure performed. The extent to which blinding of the patients can be achieved is an important part of this study; maintaining blinding of the participants is difficult because of differences in the operation time, size of skin incision, and hospital cost. Research personnel and

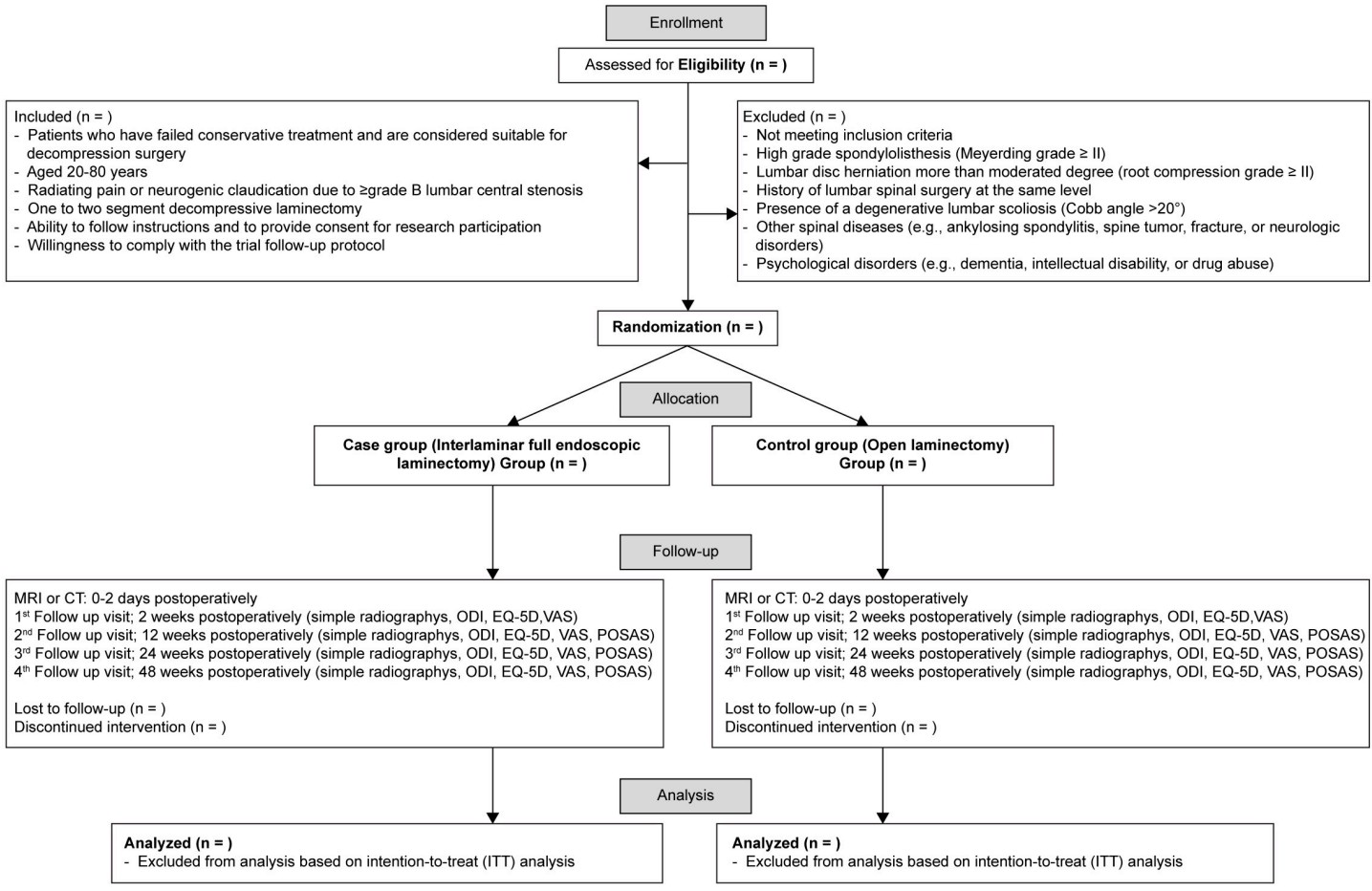

**Fig 2. Protocol flow diagram.**

healthcare providers will not disclose surgical information to patients during the study. We will determine whether the patient is blinded or not by asking the patient what surgery they had before outcome measurement. A justification will have to be submitted to the trial team by the assessor if the need for unblinding is required based on assessment findings.

## Description of the surgical interventions

The interlaminar full-endoscopic laminectomy procedure will be performed under general anesthesia with the patient in prone position, with the hips and knees flexed to adapt to the bending of the surgical table at the level of the lower lumbar spine. A 10–15-mm longitudinal skin incision will be made 1 cm lateral to the spinous process under C-arm fluoroscopic guidance. This paramedian incision will allow the endoscope to be angled appropriately to reach the contralateral side [16, 19, 28–31]. The working channel and endoscope will be introduced [16, 29], and endoscopic laminectomy and bilateral decompression will be performed through a unilateral working channel [19, 22, 28, 29, 31] (Figs 3 and 4). The decompression procedure will be performed bilaterally via unilateral access under constant saline irrigation and visual control [4, 12, 14], as shown in Fig 3B. Endoscopic bipolar cauterization will be used as necessary to achieve hemostasis [12, 14, 31]. After sufficient decompression, saline irrigation will be performed, and the endoscope will be withdrawn gradually [14, 31].

**(a)**

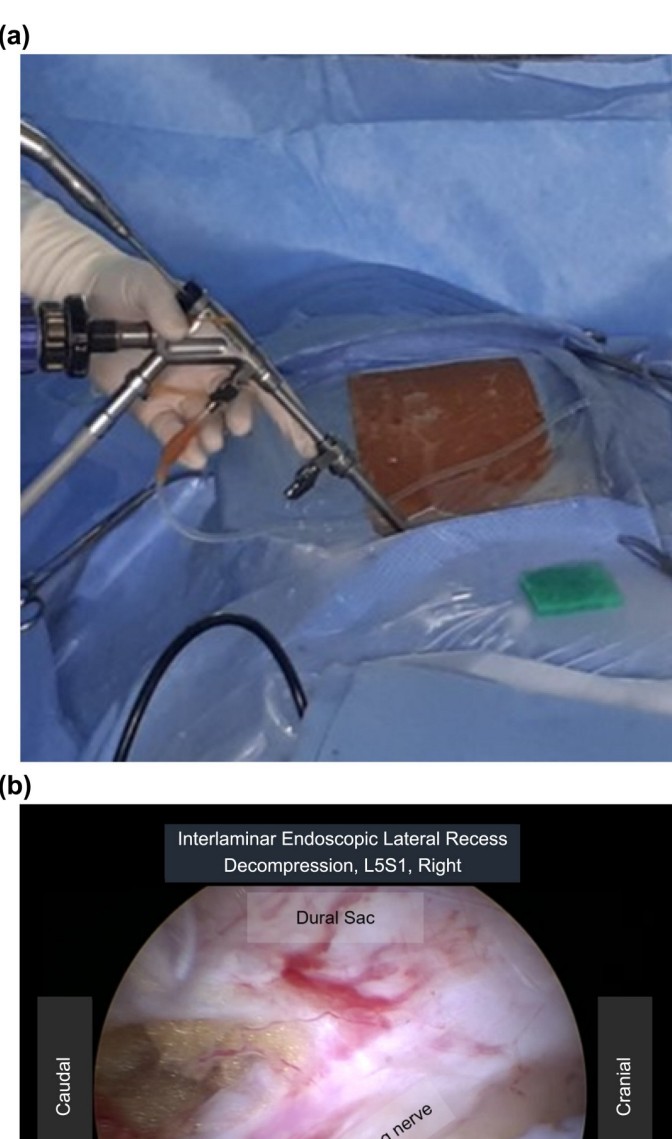

**(b)**

**Fig 3. Interlaminar full-endoscopic discectomy.** (a) The operative field of the interlaminar full-endoscopic laminectomy procedure is shown. (b) The intraoperative endoscopic view shows the laminectomy site, the decompressed thecal sac, and an individual rootlet.

The open decompressive laminectomy procedure will also be performed under general anesthesia with the patient in the prone position. A midline incision of approximately 4 cm will be made at the target level, and the paraspinal muscles will be detached using a Cobb elevator. After the parasternal muscles are retracted using a Taylor retractor and the operation field is secured, the surgery will be performed under microscopic guidance. Bilateral decompressive

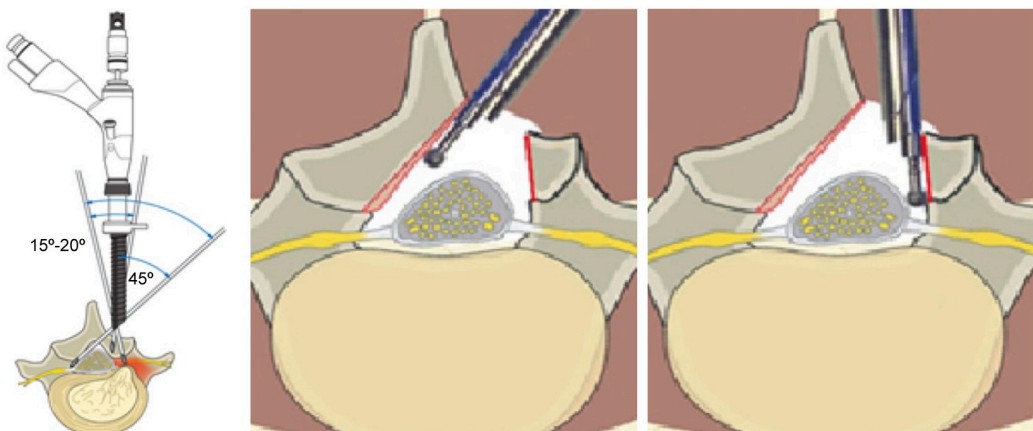

**Fig 4. Unilateral approach and bilateral decompression.** The bilateral decompression is accomplished via a unilateral access point.

laminectomy will be performed using a burr and Kerrison punch. The decompressive laminectomy and removal of the hypertrophied ligamentum flavum will be performed in the central canal and lateral recesses of the area with stenosis, resulting in the decompression of the thecal sac and any affected rootlets.

Faculty members with clinical experience of 7–12 years in both endoscopic and conventional surgeries will participate.

## Outcome measures

**Primary outcome.** The primary outcome of this trial will be the difference between the ODI score measured at baseline and that measured 12 months postoperatively [32, 33]. The ODI is the most useful and commonly used patient-reported outcome for evaluating low back pain in a hospital setting and is therefore used for evaluating the efficacy of operations. The ODI evaluates the functional level of activities of daily living for patients with low back pain based on the following 10 areas: pain intensity, personal care, lifting, walking, sitting, standing, sleeping, sex life, social life, and traveling. Each area is scored on a five-point scale, with higher scores representing greater levels of disability. The total ODI score will be divided by the total possible score and expressed as a percentage in the trial.

**Secondary outcomes.** The secondary outcomes in the trial will include patient-reported outcomes, clinical outcomes, radiographic outcomes, and adverse events. The patient-reported outcomes will include a VAS score for low back pain and pain radiating to the lower extremities. The VAS score is measured using a 10-point scale, ranging from 0 (no pain) to 10 points (severe pain). Quality of life (QOL), measured using the EQ-5D questionnaire, will also be included as a patient-reported outcome [34]. The EQ-5D questionnaire is composed of five questions, and the total EQ-5D score ranges from 0 to 1 point, with a higher score indicating a better QOL [17]. Patient satisfaction with the surgery, walking distance/time, and time required to return to activities of daily living postoperatively will also be included as patient-reported outcomes. Patient-reported outcomes will be collected from the participants at baseline and at 2 weeks and at 3, 6, and 12 months postoperatively.

The clinical outcomes of the trial will include postoperative surgical scarring, which will be measured using the Patient and Observer Scar Assessment Scale (POSAS) (version 2.0). The POSAS includes six items scored on a 10-point system, with a score of 6 points indicating

normal skin and a score of 60 points indicating the worst scar imaginable. Surgery-related variables, including operative time (minutes), duration of hospitalization (hours), postoperative drainage (mL), and postoperative creatine kinase, will also be reported as clinical outcomes.

The degree of central canal release and the canal dimension, measured using postoperative MRI or CT, will be included as radiographic outcomes in the trial. A preoperative MRI of the spine will be obtained to determine the severity of lumbar spinal stenosis and the structures causing neural compression. We will use the Schizas classification for evaluating MRI outcomes [26]. To evaluate facet joint injury, the greatest width of the medial facet on axial image with greatest width of the facet joint will be measured on preoperative and postoperative MRI examinations [35]. Other radiographic complications will be measured using simple radiographs during the follow-up period. Simple radiographs will be obtained in the anteroposterior, lateral, flexion, and extension views. Spondylolisthesis and segmental instability at the target surgical level will be scored based on these images at baseline, at 2 weeks, and at 3, 6, and 12 months postoperatively (Fig 1). The safety of each procedure will be assessed using evaluations of the occurrence and severity of adverse events and surgery-related effects. All outcomes will be collected and assessed by researchers who are blinded to the patient groups in each hospital and recorded in the eCRF system (Fig 1).

**Statistical analysis.** All statistical analyses will be performed using SAS Enterprise Guide 4 software (SAS Institute Inc., Cary, North Carolina, USA). In this non-inferiority RCT, an intention-to-treat analysis will be conducted. Data from participants who are excluded before or after surgery will not be included or replaced in the analyses to avoid the risk of selection bias.

The primary outcome (the ODI score at 12 months postoperatively) will be compared between the case and control groups. Interlaminar full-endoscopic laminectomy will be considered equivalent to open decompressive laminectomy with regard to surgical outcome if the upper and lower limits of the 95% confidence interval (CI) of the ODI score at 12 months are limited to the pre-defined equivalence limit of 12.8 points.

A linear mixed model repeated-measures analysis of variance will be performed to analyze the time-dependent changes in the secondary patient-reported and clinical outcomes (such as the VAS pain scores for the back and lower extremities, ODI, EQ-5D, and POSAS scores).

Time will be treated as a categorical variable (2 weeks and 3, 6, and 12 months postoperatively). A post-hoc test will be conducted to evaluate serial changes from baseline within each group and between the two groups at each time-point to identify significant time- and group-differences. The clinical and radiographic outcomes and adverse effects will be compared between the two groups using the chi-squared test for categorical variables and Student's t-test for continuous variables.

The distribution of the collected data will be evaluated using the Shapiro–Wilk test. Continuous variables that are normally distributed will be reported as mean and standard deviation (SD) and those that are non-normally distributed will be reported as median and interquartile range. Categorical variables will be reported as number and percentage (%).

As described above, surgeons' experience and technique of endoscopic spinal surgery may present little differences. Although there may be differences in surgical outcomes across the surgeons enrolled in this multicenter study, the differences will be verified using the Cochran–Mantel–Haenszel test. Differences between the two groups of variables used will be verified at baseline, and additional necessary analysis will be performed using analysis of covariance (ANCOVA) or mixed model analysis. The difference in age between the two groups will be verified by t-test, and ANCOVA will be performed using age as a covariate in ODI score analysis. The difference in surgical time between the two groups will be verified by t-test. If necessary, ANCOVA will be performed using age as a covariate in surgical time analysis to suggest the endoscopic technique for the treatment of elderly patients.

## Data management

Anonymous participant data will be safely and directly entered into the Korean government-created iCReaT system by the trial researchers. The iCReaT system is equipped with a web-based encryption system to prevent unauthorized access and disclosure, and only the principal investigator and designated statistical analysts can access it. The e-CRF system will be used in the trial. The iCReaT system will be managed by the clinical research coordinators at each hospital and a specialized company with extensive experience with eCRF management. Both on-site and in-house monitoring using the electronic data capture system will be conducted by designated monitoring researchers.

## Sample size determination

A total of 120 participants will be recruited, and 60 participants will be allocated to each group. According to a previous report [33], the non-inferiority margin is 12.8 points; the maximal clinically accepted ODI difference is 12.8 points with a SD of 18.8 points at 1 year after endoscopic decompressive laminectomy [36]. Assuming the equivalence limit of 12.8, an alpha value of 0.05, power of 0.90, one-sided 95% CI, and loss to follow-up of 20%, 60 participants are required in each group. Power Analysis and Sample Size software (version 15; NCSS, Kaysville, UT, USA) was used for the sample size calculations.

## Discussion

Previous studies have reported that clinical outcomes of interlaminar full-endoscopic laminectomy are not different from those of conventional laminectomy [13–15, 19–22]. However, the previous studies are limited by their retrospective or single-center designs [4, 13–15, 20, 21, 23–25]. In this trial, the quality of the evidence will be improved as adequate sample sizes, blinded assessments, and prospective registration from multiple centers will be used to reduce bias. This multicenter RCT has been designed with a high level of evidence to compare the efficacy, safety, and applicability of interlaminar full-endoscopic laminectomy and open laminectomy in patients with lumbar spinal stenosis.

The evidence collected in this trial will provide a clear conclusion regarding the efficacy and safety of interlaminar full-endoscopic laminectomy for the treatment of lumbar spinal stenosis. We anticipate that this trial will present the validity of the endoscopic technique not only for lumbar spinal stenosis, but also for various spinal surgeries, such as discectomy and spinal fusion.

## Conclusion

The outcomes of our trial will be useful to inform spinal surgeons and healthcare providers regarding the use of interlaminar full-endoscopic laminectomy as an alternative to open laminectomy for lumbar spinal stenosis with similar surgical results and less invasiveness.

## Supporting information

**S1 File. SPIRIT-checklist.**
(DOC)

**S2 File. Study protocol-the catholic univ of Korea [Eng].**
(DOCX)

**S3 File. Study protocol-the catholic univ of Korea [Kor].**
(PDF)

**S4 File. Study protocol-wiltse Memorial Hospital [Kor].**
(PDF)

**S5 File. Study protocol-wiltse Memorial Hospital [Eng].**
(DOCX)

**S6 File. Study protocol-wooridul hospital [Eng].**
(DOCX)

**S7 File. Study protocol-wooridul hospital [Kor].**
(PDF)

## Acknowledgments

We would like to thank Professors Jin-Sung Kim, Sang-Min Park, and Hyun-Jin Park for their valuable and constructive suggestions with regards to planning the trial, obtaining the funding, and developing this research.

## Author Contributions

**Conceptualization:** Jin-Sung Kim, Junseok Bae, Dong Chan Lee, Sang-Min Park, Hyun-Jin Park, Hong-Jae Lee.

**Formal analysis:** Jin-Sung Kim, Hong-Jae Lee.

**Funding acquisition:** Jin-Sung Kim, Hong-Jae Lee.

**Investigation:** Jin-Sung Kim, Junseok Bae, Dong Chan Lee, Sang-Ha Shin, Han Joong Keum, Young Soo Choi, Sang Soo Eun, Seung Ho Shin, Hyun Jin Hong, Ji Yeon Kim, Tae Hyun Kim, Woojung Lim, Junghoon Kim, Sang-Min Park, Hyun-Jin Park, Hong-Jae Lee.

**Methodology:** Jin-Sung Kim, Junseok Bae, Dong Chan Lee, Sang-Min Park, Hyun-Jin Park, Hong-Jae Lee.

**Project administration:** Jin-Sung Kim, Hong-Jae Lee.

**Supervision:** Jin-Sung Kim, Hong-Jae Lee.

**Validation:** Jin-Sung Kim, Junseok Bae, Dong Chan Lee, Sang-Ha Shin, Han Joong Keum, Young Soo Choi, Sang Soo Eun, Seung Ho Shin, Hyun Jin Hong, Ji Yeon Kim, Tae Hyun Kim, Woojung Lim, Junghoon Kim, Sang-Min Park, Hyun-Jin Park, Hong-Jae Lee.

**Writing – original draft:** Jin-Sung Kim, Hong-Jae Lee.

**Writing – review & editing:** Jin-Sung Kim, Hong-Jae Lee.

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
