## [Decision Letter · Decision Letter 0]

25 Nov 2022

PONE-D-21-32246Evaluation of the efficacy and safety of conventional and interlaminar full-endoscopic decompressive laminectomy to treat lumbar spinal stenosis (ENDO-F Trial): Protocol for a prospective, randomized, multicenter trialPLOS ONE

Dear Dr. Lee,

Thank you for submitting your manuscript to PLOS ONE. After careful consideration, we feel that it has merit but does not fully meet PLOS ONE’s publication criteria as it currently stands. Therefore, we invite you to submit a revised version of the manuscript that addresses the points raised during the review process.

We look forward to receiving your revised manuscript.

Kind regards,

Luigi Maria Cavallo

Academic Editor

PLOS ONE

Journal Requirements:

“This research is supported by a grant from the Korea Health Technology R&D Project through the Korea Health Industry Development Institute, funded by the Ministry of Health & Welfare, Republic of Korea (grant number: HC20C0163).

The funders had and will not have a role in study design, data collection and analysis, decision to publish, or preparation of the manuscript.”

3. Thank you for stating the following in the Competing Interests/Financial Disclosure* (delete as necessary) section:

“I have read the journal's policy and the authors of this manuscript have the following competing interests:

a. Consultant: Jin-Sung Kim is a consultant for RIWOSpine, GmbH, Germany, Stöckli Medical AG, Switzerland and Elliquence, LLC, USA. Jun Ho Lee is a consultant for RIWOSpine, GmbH, Germany. Junseok Bae is a consultant for Joimax, GmbH, Germany. The other authors declare that they have no competing interests.

b. Stock/Shareholder: nothing

c. Speakers’ Bureau: nothing”

We note that one or more of the authors are employed by a commercial company: RIWOSpine, GmbH, Germany, Stöckli Medical AG, Switzerland and Elliquence, LLC, USA

Reviewers' comments:

Reviewer's Responses to Questions

**Comments to the Author**

1. Does the manuscript provide a valid rationale for the proposed study, with clearly identified and justified research questions?

Reviewer #1: Yes

2. Is the protocol technically sound and planned in a manner that will lead to a meaningful outcome and allow testing the stated hypotheses?

Reviewer #1: Yes

3. Is the methodology feasible and described in sufficient detail to allow the work to be replicable?

Reviewer #1: Yes

4. Have the authors described where all data underlying the findings will be made available when the study is complete?

Reviewer #1: Yes

5. Is the manuscript presented in an intelligible fashion and written in standard English?

Reviewer #1: Yes

6. Review Comments to the Author

You may also provide optional suggestions and comments to authors that they might find helpful in planning their study.

Reviewer #1: We have carefully examined this Manuscript presented for Plos One.

The authors present a randomized controlled trial to compare safety and efficacy of interlaminar full-endoscopic laminectomy and open decompressive laminectomy.

We kindly ask the Authors to accept our criticism:

● there is no mention about the surgeons’ experience with the endoscopic spinal technique. Please consider this aspect

● it is necessary to define the outcomes in regard to the age, underlining the “statistical difference” between the two groups.

● it is interesting to define the surgical time difference between the two groups, in order to suggest the endoscopic technique for the treatment of the elderly patients.

7. PLOS authors have the option to publish the peer review history of their article (what does this mean?). If published, this will include your full peer review and any attached files.

Reviewer #1: No

---

## [Author Response · Author response to Decision Letter 0]

10 Jan 2023

Response to Reviewers

Comments from Reviewer #1

Comment 1: there is no mention about the surgeons’ experience with the endoscopic spinal technique. Please consider this aspect.

Response: We would like to thank Reviewer 1 for the time and efforts in reviewing our manuscript and for providing comments, which have considerably helped us improve our manuscript. We have made revisions based on the comments and have provided our point-by-point responses below. The revisions are indicated using track changes in the revised manuscript. We hope that our responses and revisions appropriately address the reviewer’s comments. 

We agree with the suggestion to mention the surgeon’s experience and have included the following information in the revised manuscript.

“Faculty members with clinical experience of 7–12 years in both endoscopic and conventional surgeries will participate.” (Lines 208–209 of the revised manuscript) 

Specifically, these experts perform 5–10 endoscopic surgeries per month. 

“As described above, surgeons’ experience and technique of endoscopic spinal surgery may present little differences. Although there may be differences in surgical outcomes across the surgeons enrolled in this multicenter study, the differences will be verified using the Cochran–Mantel–Haenszel test.” 

We have provided this information in the revised manuscript (Lines 279–282).

Comment 2: it is necessary to define the outcomes in regard to the age, underlining the “statistical difference” between the two groups.

Response: We agree with the reviewer’s comment. We have provided the following information in the revised manuscript (Lines 282–285).

“Differences between the two groups of variables used will be verified at baseline, and additional necessary analysis will be performed using analysis of covariance (ANCOVA) or mixed model analysis. The difference in age between the two groups will be verified by t-test, and ANCOVA will be performed using age as a covariate in ODI score analysis.” 

Comment 3: it is interesting to define the surgical time difference between the two groups, in order to suggest the endoscopic technique for the treatment of the elderly patients.

Response: We agree with the reviewer’s comment. We have provided the following information in the revised manuscript (Lines 286–288).

“The difference in surgical time between the two groups will be verified by t-test. If necessary, ANCOVA will be performed using age as a covariate in surgical time analysis to suggest the endoscopic technique for the treatment of elderly patients.”

---

## [Decision Letter · Decision Letter 1]

21 Mar 2023

Evaluation of the efficacy and safety of conventional and interlaminar full-endoscopic decompressive laminectomy to treat lumbar spinal stenosis (ENDO-F Trial): Protocol for a prospective, randomized, multicenter trial

PONE-D-21-32246R1

Dear Dr. Lee,

We’re pleased to inform you that your manuscript has been judged scientifically suitable for publication and will be formally accepted for publication once it meets all outstanding technical requirements.

Kind regards,

Luigi Maria Cavallo

Academic Editor

PLOS ONE

Additional Editor Comments (optional):

Reviewers' comments:

Reviewer's Responses to Questions

**Comments to the Author**

1. Does the manuscript provide a valid rationale for the proposed study, with clearly identified and justified research questions?

Reviewer #1: Yes

2. Is the protocol technically sound and planned in a manner that will lead to a meaningful outcome and allow testing the stated hypotheses?

Reviewer #1: Yes

3. Is the methodology feasible and described in sufficient detail to allow the work to be replicable?

Reviewer #1: Yes

4. Have the authors described where all data underlying the findings will be made available when the study is complete?

Reviewer #1: Yes

5. Is the manuscript presented in an intelligible fashion and written in standard English?

Reviewer #1: Yes

6. Review Comments to the Author

You may also provide optional suggestions and comments to authors that they might find helpful in planning their study.

Reviewer #1: The paper could be accepted in this form.

Authors addressed the revisions previous required and the paper is very interesting

7. PLOS authors have the option to publish the peer review history of their article (what does this mean?). If published, this will include your full peer review and any attached files.

Reviewer #1: **Yes: **Teresa Somma

---

## [Editor Report · Acceptance letter]

28 Mar 2023

PONE-D-21-32246R1 

Evaluation of the efficacy and safety of conventional and interlaminar full-endoscopic decompressive laminectomy to treat lumbar spinal stenosis (ENDO-F Trial): Protocol for a prospective, randomized, multicenter trial 

Dear Dr. Lee:

I'm pleased to inform you that your manuscript has been deemed suitable for publication in PLOS ONE. Congratulations! Your manuscript is now with our production department. 

Kind regards, 

on behalf of

Dr. Luigi Maria Cavallo 

Academic Editor

PLOS ONE